# Spatial and Channel Aggregation Network for Lightweight Image Super-Resolution

**DOI:** 10.3390/s23198213

**Published:** 2023-10-01

**Authors:** Xianyu Wu, Linze Zuo, Feng Huang

**Affiliations:** College of Mechanical Engineering and Automation, Fuzhou University, Fuzhou 350108, China; xwu@fzu.edu.cn (X.W.); linzezuo@gmail.com (L.Z.)

**Keywords:** lightweight image super-resolution, large kernel convolution, peak signal-to-noise ratio (PSNR) metric

## Abstract

Advanced deep learning-based Single Image Super-Resolution (SISR) techniques are designed to restore high-frequency image details and enhance imaging resolution through the use of rapid and lightweight network architectures. Existing SISR methodologies face the challenge of striking a balance between performance and computational costs, which hinders the practical application of SISR methods. In response to this challenge, the present study introduces a lightweight network known as the Spatial and Channel Aggregation Network (SCAN), designed to excel in image super-resolution (SR) tasks. SCAN is the first SISR method to employ large-kernel convolutions combined with feature reduction operations. This design enables the network to focus more on challenging intermediate-level information extraction, leading to improved performance and efficiency of the network. Additionally, an innovative 9 × 9 large kernel convolution was introduced to further expand the receptive field. The proposed SCAN method outperforms state-of-the-art lightweight SISR methods on benchmark datasets with a 0.13 dB improvement in peak signal-to-noise ratio (PSNR) and a 0.0013 increase in structural similarity (SSIM). Moreover, on remote sensing datasets, SCAN achieves a 0.4 dB improvement in PSNR and a 0.0033 increase in SSIM.

## 1. Introduction

Single Image Super-Resolution (SISR) refers to the technique of restoring Low-Resolution (LR) images to High-Resolution (HR) clear images. SISR not only improves the perceptual quality of images but also helps to enhance the performance of other computer vision tasks such as object detection and image denoising [1,2,3]. As a result, it has attracted wide attention from researchers.

Since the success of the first SISR method based on Convolutional Neural Networks (CNN) [4], many excellent CNN-based SISR methods have emerged [5,6,7,8,9,10,11], many of which are built upon classic architectures such as residual networks [12], dense networks [13], hybrid network [14] and visual attention mechanisms [15,16,17], among others.

Edge computing devices have made remarkable strides [18,19], enabling the deployment of super-resolution algorithms. However, the existing algorithms, due to their high computational complexity, struggle to strike a balance between performance and speed on edge devices. Therefore, there is a pressing need to develop SISR methods that are lightweight and efficient, ensuring their viability for resource-constrained devices.

Accordingly, there have been many outstanding works on lightweight SISR networks [20,21,22,23,24,25,26,27], most of which employ more compact network architectures and utilize ingenious lightweight strategies. These lightweight strategies include the use of group convolutions [28], depth-wise separable convolutions [29], dilated convolutions [30], and cross convolution [31] to replace regular convolutions. In addition, there are also lightweight strategies such as neural architecture search [32,33], structural reparameterization [34], efficient attention mechanisms [35] and so on. Despite the increasing availability of lightweight SISR networks, their performance is often severely com-promised, making it difficult for them to meet the demands of complex practical applications. There remains room for improvement in the field of lightweight SISR.

Recent research suggests that the remarkable performance of ViT is primarily at-tributed to its macro architecture [36,37]. By utilizing advanced training configurations and adopting ViT-inspired architectural enhancements, CNN can achieve performance on par with, or even surpass, that of ViT, especially when employing large kernel convolutions [38,39]. Following that, Yu et al. [40] present MetaFormer as a general architecture abstracted from transformers, as shown in Figure 1, MetaFormer architecture consists of normalization layers, Spatial Mixture (SM) modules, and Channel Mixture (CM) modules. Further investigation by Yu et al. [41] reveals that a pure CNN network with the MetaFormer architecture is more efficient compared to the ViT-based network in image classification task. The Multi-scale Attention Network (MAN) [42] is an exceptional work of SISR that employs the MetaFormer architecture and large kernel convolutions. MAN employ large convolutional kernels in their spatial and channel mixture modules to extract information, resulting in more efficient performance compared to ViT-based SR methods.

Furthermore, Deng et al. [43] found that deep neural networks have a tendency to encode interactions that are either too complex or too simple, rather than interactions with a moderate level of complexity. MAN also suffer from this drawback. A recent work focusing on the Multi-Order Gated Aggregation Network (MogaNet) [44] proposed the use of feature reduction operations, which compel the network to focus more on challenging intermediate-level information, and achieved excellent results in image classification tasks.

Inspired by the Multi-Order Gated Aggregation Network (MogaNet) [44], we introduced the strategy of large-kernel convolutions coupled with feature reduction in the field of super-resolution for the first time and, based on this design, developed the Triple-Scale Spatial Aggregation Attention Module to aggregate multi-scale information. Building upon the MetaFormer architecture, we further proposed a Spatial and Channel Aggregation Block (SCAB) to aggregate multi-order spatial and channel information. Furthermore, an ingenious introduction of a 9 × 9 large kernel convolutional layer is made at the end of the SCAB module to further expand the receptive field.

As shown in Figure 2, the proposed SCAN can aggregate more contextual information compared to the light versions of MAN [42] and Image Restoration Using Swin Transformer (SwinIR) [45].

The main contributions of this work can be summarized as follows:(1)The Triple-Scale Spatial Aggregation (TSSA) attention module was innovatively introduced for the first time, enabling the aggregation of triple-scale spatial information.(2)The Spatial and Channel Aggregation Block (SCAB) is innovatively introduced for the first time, capable of aggregating both multi-scale spatial and channel information.(3)The Spatial and Channel Aggregation Network (SCAN), a lightweight and efficient pure CNN-based SISR network model that combines the advantages of both CNN and Transformer is proposed.(4)Quantitative and qualitative evaluations are conducted on benchmark datasets and remote sensing datasets to investigate the proposed SCAN. As shown in Figure 3, the proposed SCAN achieves a good trade-off between model performance and complexity.

## 2. Related Work

Classical Deep Learning-based SISR models. With the rapid development of deep learning techniques, researchers have been actively exploring and studying deep learning-based SISR methods. Compared to traditional approaches, deep learning-based methods can extract more expressive image features from the dataset and adaptively learn the mapping relationship between low-resolution and high-resolution images. Consequently, they have achieved remarkable breakthroughs in this field.

Dong et al. [4] first introduced the application of deep learning in the field of SISR by proposing the pioneering CNN-based SISR network model called Super-Resolution CNN (SRCNN), which utilizes three convolutional layers to achieve an end-to-end mapping between low-resolution and high-resolution images, resulting in superior performance compared to traditional methods. Since then, numerous out-standing deep learning-based SISR methods have emerged. Kim et al. [5] introduced Residual Network (ResNet) [12] into the field of SISR inspired by the deep convolutional neural network VGG-net, and proposed the Very Deep SR (VDSR) with 20 weight layers. Kim et al. [48] introduced Recurrent Neural Network (RNN) into SISR for the first time and proposed the Deep Recursive Convolutional Network (DRCN) with up to 16 recursive layers, combining residual learning to control parameter count and address the overfitting issues associated with increasing network depth. Tong et al. [8] pioneered the application of Dense Convolutional Network (DenseNet) [13] in SR and proposed the SR DenseNet network (SRDenseNet) model, which effectively fuses low and high-level features through dense skip connections and enhances the reconstruction of image details using deconvolutional layers. Lim et al. [6] introduced the Enhanced Deep SR (EDSR) network, which removes Batch Normalization (BN) layers from the SRResNet [49] network and incorporates techniques like residual scaling. Their work demonstrated a remarkable enhancement in the quality of image reconstruction. Addressing the oversight of high-frequency information recovery in existing deep learning-based SR methods, Wu et al. [14] introduced a new convolutional block, known as the Spatial-Frequency Hybrid Convolution Block (SFBlock). This block is engineered to extract features from both the spatial and frequency domains. It enhances the capturing of high-frequency details, while simultaneously preserving low-frequency information.

The attention mechanism [50], by assigning different weights to image features based on their importance, enables the network to prioritize crucial information with higher weights. As a result, it is widely employed in various visual tasks to enhance performance and focus on important information. Zhang et al. [11] introduced the attention mechanism into SR for the first time and proposed the Residual Channel Attention Network (RCAN), which utilizes channel attention to adaptively recalibrate the features of each channel based on their interdependencies. This approach enhances the network’s representation capability by learning more informative channel features. Subsequently, Dai et al. [16] pointed out that the channel attention mechanism introduced in RCAN only utilizes the first-order statistics of features through global average pooling, which neglects higher-order statistics and hinders the network’s discriminative ability. They proposed a second-order attention mechanism and developed the Second-order Attention Network (SAN), which achieved better performance than RCAN. Wu et al. [17] introduced a novel Feedback Pyramid Attention Network (FPAN) for SISR. By leveraging the unique feedback connection structure proposed in their work, the FPAN is capable of enhancing the representative capacity of the network while facilitating information flow within it. Furthermore, it exhibits a proficiency in capturing long-range spatial context information across multiple scales.

Recently, the transformer [39] has made significant breakthroughs in the field of computer vision. Inspired by this, Chen et al. [51] proposed a pre-trained network model called Image Processing Transformer (IPT) for handling various low-level computer vision tasks such as SR, denoising, and rain removal. The IPT model proves to be effective in performing the desired tasks, outperforming most existing methods across different tasks. Liang et al. [45] introduced Image Restoration Using Swin Transformer (SwinIR), a high-quality image SR model based on the Swin Transformer [30] architecture. By leveraging the window-based attention mechanism of Swin Transformers, SwinIR effectively handles spatial relationships in images and learns the mapping between low-resolution and high-resolution images, resulting in superior image SR performance.

Guo et al. [39] proposed the Large Kernel Attention (LKA) module and developed the Visual Attention Network (VAN), which outperforms state-of-the-art VIT-based networks in multiple visual tasks. Inspired by LKA, Li et al. [42] proposed the Multi-Scale Large Kernel Attention (MLKA) and developed the Multi-Scale Large Kernel Attention Net-work (MAN) for SISR. MAN outperforms SwinIR with improved performance and reduced computational cost, thanks to the integration of MLKA.

Although these classic SISR methods have achieved impressive performance, they often come with a significant computational cost, making it challenging to apply them in resource-constrained scenarios.

**Lightweight SISR models.** To facilitate the practical application of SISR methods, researchers have also proposed numerous lightweight SISR methods. Dong et al. [20] introduced Fast SRCNN (FSRCNN), which replaces the pre-upsampling model frame-work in SRCNN with a post-upsampling model framework. By utilizing a deconvolution layer at the end of the network for upsampling, FSRCNN effectively addresses the high computational complexity of SRCNN. Shi et al. [52] proposed Efficient Sub-Pixel CNN (ESPCN), which, like FSRCNN, adopts a post-upsampling model framework. However, ESPCN employs sub-pixel convolutional layers for image upsampling, resulting in superior reconstruction performance compared to the FSRCNN network model.

Ahn et al. [21] introduced the Cascading Residual Network (CARN) which enhances network efficiency through the utilization of group convolutions and parameter sharing blocks. Hui et al. [22] also employed the strategy of group convolutions and introduced the Information Distillation block to construct the Information Distillation Network (IDN), which further improves the quality of image reconstruction while enhancing the speed. Building upon IDN, Hui et al. [23] improved the information distillation blocks and designed the Information Multi-Distillation Block (IMDB) for constructing a lightweight Information Multi-Distillation Network (IMDN), which achieves higher efficiency performance compared to IDN. Li et al. [53] proposed the linearly assembled pixel-adaptive regression network (LAPAR), which transforms the problem of learning the LR-to-HR image mapping into a linear regression task based on multiple predefined filter dictionaries. This approach achieves good performance while maintaining speed. Luo et al. [54] introduced the lattice block network (LatticeNet), which utilizes multiple cascaded lattice blocks based on a lattice filter bank, as well as backward feature fusion strategy. Inspired by edge detection methodologies, Liu et al. [31] developed a novel cross convolution approach that enables more effective exploration of the structural information of the features.

Song et al. [26] proposed an efficient residual dense block search algorithm for image SR. It employs a genetic algorithm to search for efficient SR network structures. Huang et al. [32] introduced a lightweight image SR method with a fully differentiable neural architecture search (DLSR), which incorporates a fully differentiable neural architecture search. A key innovation of their work lies in the creation of a cell-level and network-level search space, which enables the discovery of optimal lightweight models. Zhang et al. [25] introduced the reparameterization concept to SISR and proposed edge-oriented convolution block for real-time SR (ECBSR). During training, they utilized a multi-branch module to enhance the model’s performance. During inference, they transformed the multi-branch module into a single-branch structure to improve runtime speed while maintaining performance. Zhu et al. [35] introduced a light-weight SISR network known as Expectation-Maximization Attention SR (EMASRN). The distinctive aspect of their work is the incorporation of a novel high-resolution expectation-maximization attention mechanism, which effectively captures long-range dependencies in high-resolution feature maps.

Although the aforementioned methods achieve high processing speeds, they often struggle to achieve satisfactory image reconstruction quality. In recent lightweight SISR research, there has been a growing trend to combine ViT and LKA to achieve improved reconstruction quality.

Lu et al. [55] proposed the Efficient SR Transformer (ESRT), a hybrid network composed of a CNN backbone and a transformer backbone to address the significant computational cost and high GPU memory consumption of transformers. Sun et al. [56] introduced ShuffleMixer, which employs deep convolutions with large kernel sizes to aggregate spatial information from large regions. Sun et al. [57] introduced the Spatially Adaptive Feature Modulation Network (SAFMN), a CNN SR network based on the ViT architecture, which achieved a balance between performance and model complexity. Wu et al. [58] proposed TCSR, which introduces the neighborhood attention module to achieve more efficient performance than LKA.

While these lightweight methods have achieved a good balance between model complexity and performance, there is still room for improvement.

## 3. Proposed Method

### 3.1. Network Architecture

As illustrated in Figure 4, the proposed SCAN consists of the following three components: the Shallow Feature (SF) extraction module, the Deep Feature (DF) extraction module based on cascaded Spatial and Channel Aggregation Groups (SCAG), and the high-quality image reconstruction module.

**Shallow feature extraction module (SF):** given an input Low-Resolution (LR) image $I_{LR}$ ∈ $R^{3\times H\times W}$, where H and W are the height and width of the LR image. The shallow feature extraction module was applied, denoted by ${ƒ}_{SF}$(⋅), which consists of only a single 3 × 3 convolution, to extract the shallow feature $F_{p}$∈$R^{C\times H\times W}$. The process is expressed as follows:(1)$$F_{p}={ƒ}_{SF}(I_{LR})$$

**Deep feature extraction module (DF):** then the shallow features are sent to the deep feature extraction module to extract deeper and more abstract high-level features. The process is expressed as follows:(2)$$F_{n}=DWD\;Conv_{9\times 9,d=4}(F_{n-1}).$$

Where $F_{r}$ denotes the deep feature maps. ${ƒ}_{DF}$(⋅) denotes the deep feature extraction module, which consists of multiple cascaded SCAGs and a single 9 × 9 depth-wise-dilated convolutional layer with dilation ratios d = 4. More specifically, intermediate features $F_{1}$, $F_{2}$, … $F_{n}$ are extracted step by step, as shown in the following formula:(3)$$\begin{array}{c} F_{i}={ƒ}_{SCAG_{i}}(F_{i-1}),\;i=1,2,\ldots ,n\\ F_{n}=DWD\;Conv_{9\times 9,\;d=4}(F_{n-1}). \end{array}$$where ${ƒ}_{SCAG_{i}}$ (⋅) denotes the *i*th SCAG and $DWD\;Conv_{9\times 9,d=4}$ (⋅) denotes 9 × 9 depth-wise-dilated convolutional layer with dilation ratios d = 4. *n* denotes the number of SCAG.

**Image reconstruction module:** $F_{r}$ and $F_{p}$ are sent to the image reconstruction module to complete the super resolution reconstruction of the image. The process can be described as below:(4)$$I_{SR}={ƒ}_{RC}(F_{p}+F_{r})+Bicubic(I_{LR}).$$
where ${ƒ}_{RC}$ denotes the up-sampling module, which consists of a pixel shuffle operation and a single 3 × 3 convolution. Bicubic(⋅) denotes bicubic interpolation up-sampling operation. Incorporating interpolation at this juncture serves to enhance network performance and expedite network convergence.

### 3.2. Spatial and Channel Aggregation Groups (SCAG)

As discussed in the first section, neural networks utilizing Metaformer [40] architectures similar to ViT have emerged with tremendous potential. Flowing to Metaformer, we propose a spatial and channel aggregation module which used a Metaformer-style design.

As shown in Figure 5, SCAG consists of multiple cascaded spatial aggregation and channel aggregation blocks (SCAB) and a single 9 × 9 depth-wise-dilated convolutional layer with dilation ratios d = 4.

**Spatial aggregation and channel aggregation block (SCAB):** SCAB consists of the following three components: the triple-scale spatial aggregation attention (TSSA) module, the channel aggregation (CA) module, and the layer normalization layer.

Given the input feature X, the whole process of SCAB is formulated as:(5)$$\begin{array}{c} N=LN(X)\\ X_{1}=X+TSSA(N)\\ N_{1}=LN(X_{1})\\ X_{2}=X_{1}+CA(N_{1}) \end{array}$$
where LN (·) are layer normalization and is employed to enhance network training stability, accelerate convergence speed, and improve model generalization performance. TSGA (·) and CA (·) denote the triple-scale gated aggregation attention (TSGA) module and the channel aggregation (CA) module, which will be introduced in the next section.

#### 3.2.1. Triple-Scale Spatial Aggregation (TSSA) Attention Module

As mentioned in the first section, methods based on large kernel convolutions have gradually surpassed ViT-based methods in some computer vision domains. However, there still exists a bottleneck in representation with the following current methods: deep neural networks have a tendency to encode interactions that are either too complex or too simple, rather than interactions with a moderate level of complexity.

In MogaNet [44], the authors innovatively introduced feature decomposition operations to encourage the network to pay more attention to intermediate-level information that is often overlooked by deep neural networks. Inspired by MogaNet [44] and CNN methods that employ large kernel convolutions [39,42], we propose triple-scale spatial aggregation attention module (TSSA) to aggregate triple-scale context information. As shown in Figure 6, the whole process of TSSA is formulated as:(6)$$\begin{array}{c} Y=FD(X)\\ Z=TSGA(Y) \end{array}$$
where FD (·) indicates a Feature Decomposition (FD) module used to eliminate redundant feature interactions. TSGA (·) is a Triple-Scale Gated Aggregation (TSGA) module used to aggregate triple-scale contextual information.

**The Feature Decomposition (FD) module** can be formulated as:(7)$$\begin{array}{c} Y=Conv_{1\times 1}(X).\\ Z=GELU(Y+\gamma_{S}⊗(Y-GAP(Y))) \end{array}$$
where $Conv_{1\times 1}$(·) and GAP(·) are 1 × 1 convolutional layer and global average pooling layer which can extract common local texture and complex global shape, separately. Y−GAP(Y) can increase the impact of mid-level information. $\gamma_{S}$ ∈ $R^{C\times 1}$ denotes a scaling factor initialized as zeros, which can increases spatial feature diversities by re-weighting. GELU indicates gaussian error linear unit [59], a high-performing neural network activation function, used for channel information gathering and redistribution.

**Triple-Scale Gated Aggregation (TSGA) module** can be formulated as:(8)$$\begin{array}{c} Y=DWConv_{5\times 5}(X)\\ Y_{l},Y_{m},Y_{h}=Split(Y)\\ Y_{l}=DWDConv_{5\times 5,d=2}(Y_{l})\\ Y_{m}=DWDConv_{7\times 7,d=3}(Y_{m})\\ Y_{h}=DWDConv_{9\times 9,d=4}(Y_{h})\\ Y=Concat(Y_{l},Y_{m},Y_{h})\\ Z=SiLU(Conv_{1\times 1}(X)⊗SiLU(Conv_{1\times 1}(Y))). \end{array}$$
where $DWConv_{5\times 5}$(·) are 5 × 5 depth-wise convolutional layer. Then, by Split(·) the output Y is equally divided into $Y_{l}\in R^{C_{l}\times HW},Y_{m}\in R^{C_{m}\times HW},Y_{h}\in R^{C_{h}\times HW}$ along the channel dimension, where $C_{l}=C_{m}=C_{h}=C/3$. To extract triple scale feature, $Y_{l},Y_{m},Y_{h}$ are sent to the 5 × 5 depth-wise-dilated convolutional layer with a dilation ratio of d = 2, 7 × 7, the depth-wise-dilated convolutional layer with a dilation ratio of d = 3, 9 × 9, and the depth-wise-dilated convolutional layer with a dilation ratio of d = 4, separately. Then, the output $Y_{l},Y_{m},Y_{h}$ are concatenated to form triple-scale contexts by Concat(·). SiLU are sigmoid-weighted linear units [60], which possesses both the gating effects of Sigmoid and the stable training properties. $Conv_{1\times 1}$ is 1 × 1 convolutional layer. ⊗ is element-wise multiplication. Finally, spatial gates are leveraged to learn more local information.

#### 3.2.2. Channel Aggregation (CA) Module

Metaformer-style architectures, as illustrated in first section, perform Channel Mixing (CM) usually by 2-layer channel MLP or MLP equipped with 3 × 3 depth-wise convolution [61,62,63]. The suboptimal efficiency of traditional MLPs can be attributed to redundant channels. To address this issue, a lightweight channel aggregation module was introduced, which was inspired by MogaNet. As illustrated in Figure 7, the Channel Aggregation (CA) module is formulated as:(9)$$\begin{array}{c} Y=GELU(DWConv_{3\times 3}(Conv_{1\times 1}(X)))\\ Z=Conv_{1\times 1}(MSFR(Y)). \end{array}$$
where $Conv_{1\times 1}$ (·) indicates a 1 × 1 convolutional layer, $DWConv_{3\times 3}$(·) indicates 3 × 3 depth-wise convolutional layer, GELU(·) indicates gaussian error linear unit [62], a high-performing neural network activation function.

MSFR(·) indicates multi-scale feature reallocation module, which is realized through a channel-reducing projection and gaussian error linear unit [62] to aggregate and redistribute channel-specific information.

**The Multi-Scale Feature Reallocation (MSFR) module** is formulated as:(10)$$MSFR(X)=X+\gamma_{c}⊗(X-GELU(XW_{r})).$$

$W_{r}$ indicates the channel-reducing projection, which is formulated as:(11)$$R^{C\times HW}\to R^{1\times HW}.$$

The emphasis on mid-level features is enhanced by performing the $X-GELU(XW_{r})$ operation, which subtracts the global channel information. This operation effectively reduces the influence of global channel statistics and enables a stronger focus on mid-level information.$\gamma_{c}$ indicates the channel-wise scaling factor initialized as zeros, which increases channel feature diversities.

## 4. Experiments

### 4.1. Experimental Setup

**Datasets and Evaluation Metrics.** The training images consist of 2650 images fromFlickr2K [6] and 800 images from DIV2K [64]. We evaluated our models on the following widely used benchmark datasets: Set5 [47], Set14 [65], BSD100 [66], Urban100 [67], and Manga109 [68]. The commonly used data augmentation methods are applied in the training dataset. Specifically, we used a random combination of random rotations of 0°, 90°, 180°, 270° and horizontal flipping for data augmentation. The average Peak Signal-to-Noise Ratio (PSNR) and the Structural Similarity (SSIM) on the luminance (Y) channel are used as the evaluation metrics.

**Implementation Details.** For a more comprehensive evaluation of the proposed methods, two different versions of SCAN were trained to resolve the lightweight SR tasks under different complexity. 1/6 SCAGs and 5/4 SCABs were stacked, and the channel width was set to 48/60 in the corresponding tiny/light SCAN.

**Training Details.** The model was trained using the Adam optimizer [69] with β1 = 0.9 and β2 = 0.99. The learning rate was initialized as 5×$10^{-4}$ and scheduled by cosine annealing learning during the whole 1×$10^{6}$ training iterations. For the ablation studies, we trained all models in 4×$10^{5}$ iterations. The weight of the exponential moving average (EMA) [70] was set to 0.999. Only the L1 loss was used to optimize the model. The patch size/batch size was set to 192 × 192/64 and 192 × 192/32 in the corresponding tiny/light SCAN.

### 4.2. Comparison with SCAN-Tiny SR Method

**Quantitative comparisons.** To evaluate the performance of the proposed SCAN-tiny, a comparison was made with state-of-the-art tiny SR methods with a parameter count of around 200 k, including Bicubic, SRCNN [4], FSRCNN [20], ShuffleMixer-tiny [56], ECBSR [25], LAPAR-B [53], MAN-tiny [42], and SAFMN [57]. Table 1 shows the quantitative comparison on benchmark datasets for the upscale factors of ×2, ×3 and ×4. The number of parameters (Params) and Floating-point Operations (FLOPs) are also provided, calculated on the 1280 × 720 output. Benefiting from its simple and efficient architecture, the proposed SCAN-tiny achieved comparable performance with fewer parameters and memory consumption.

As shown in Table 1, the proposed SCAN surpassed all methods with parameter counts less than 200 k. Specifically, compared to MAN-tiny ×4, the proposed SCAN-tiny ×4 achieves average 0.1 dB PSNR gain and 0.00216 SSIM gain on five benchmark datasets, with parameter and computational complexity almost equivalent to MAN-tiny ×4. Similarly, compared to ShuffleMixer-tiny ×4, the proposed SCAN-tiny ×4 achieved on average a 0.244 dB PSNR gain and a 0.00634 SSIM gain on the five benchmark datasets, with only slightly higher parameter and computational complexity than ShuffleMixer-tiny.

In addition to networks with fewer than 200 k parameters, comparisons were also made with many networks having more than 200 k parameters. Specifically, the proposed SCAN-tiny ×4 uses 53% parameter and 18% computational complexity of LAPAR-B ×4, achieves average 0.168 dB PSNR gain and 0.00368 SSIM gain on five benchmark datasets. Moreover, SAFMN, which ranks in the Top 3 for model complexity in the NTIRE2023 ESR Challenge, is also listed. The proposed SCAN-tiny ×4 demonstrates comparable performance to SAFMN ×4, while utilizing only 69% of the parameters and 68% of the computational complexity.

These results validate that the proposed SCAN achieves superior performance with fewer parameters, significantly enhancing computational efficiency.

**Qualitative comparisons.** In addition to the quantitative evaluations, a visual comparison is presented between the SCAN-tiny and six state-of-the-art tiny SR methods, including Bicubic, SRCNN [4], FSRCNN [20], ECBSR [25], LAPAR-B [53], and MAN-tiny [42]. Figure 8 presents a visual comparison of state-of-the-art methods on the Urban100 (×4) and Set14 (×4) datasets for upscale factors of ×4. The image within the red box is cropped and zoomed in on.

For img096 from Urban100, none of the six methods compared to the proposed SCAN were able to recover realistically sharp edges, and the resulting images exhibit blurriness and artifacts. The proposed SCAN reconstructed images closely resemble the HR images. Similarly, for barbara from Set14, only the proposed SCAN is capable of restoring authentic and clear images of books. These visual results demonstrate the information extraction capabilities of the proposed SCAN.

### 4.3. Comparison with Light SR Method

**Quantitative comparisons.** To assess the performance of the proposed SCAN, a comparison is conducted with state-of-the-art lightweight SR methods with a parameter count of approximately 1000 k, including CARN [21], LatticeNet [54], LAPAR-A [53], IMDN [23], ESRT [55], SwinIR-light [45], TCSR-L [58] and MAN-light [42]. Table 2 shows the quantitative comparison on benchmark datasets for upscale factors of ×2, ×3 and ×4. The number of parameters (Params) and floating-point operations (FLOPs) are provided, calculated on the 1280 × 720 output.

Thanks to its straightforward and efficient architecture, the proposed SCAN model achieves comparable performance while utilizing fewer parameters and consuming less memory. As shown in Table 2, the proposed approach surpasses all methods with a parameter count of around 1000 k.

Specifically, compared with several state-of-the-art transformer-based methods such as ESRT and SwinIR-light. The proposed SCAN ×4 uses 77% floating-point operations of ESRT ×4, while achieving a 0.36 dB PSNR gain and 0.00624 SSIM gain on five benchmark datasets. Moreover, compared to SwinIR-light ×4, the proposed SCAN ×4 achieves an average of 0.204 dB PSNR gain and 0.00324 SSIM gain on five benchmark datasets, with parameters and computational complexity almost equivalent to SwinIR-light ×4.

In addition to transformer-based methods, many CNN-based methods were also compared. Specifically, the proposed SCAN ×4 uses 88% parameter and 56% computational complexity of TCSR-L ×4, achieves an average of 0.096 dB PSNR gain and 0.00172 SSIM gain on the five benchmark datasets. Moreover, compared to MAN-light ×4, the proposed SCAN ×4 achieves an average of 0.085 dB PSNR gain and 0.00016 SSIM gain on the five benchmark datasets, with only slightly higher parameters and computational complexity than MAN-light ×4.

Compared to CNN-based and transformer-based methods, the proposed SCAN achieves superior performance with lower computational complexity, demonstrating the superiority of the SCAN approach.

**Qualitative comparisons**. In addition to the quantitative evaluations, a visual com parison of the proposed SCAN and six state-of-the-art light SR methods is presented, including Bicubic, LAPAR-A [53], IMDN [23], ESRT [55], SwinIR-light [45] and MAN-light [42]. Figure 9 presents a visual comparison of state-of-the-art methods on the Urban100 (×4) and Set14 (×4) datasets for up-scale factors of ×4. The image within the red box is cropped and zoomed in on.

For img092 from Urban100, the proposed SCAN method can restore images that are nearly indistinguishable from the HR, while other methods lack this capability, whereas other methods fail to achieve this level of quality, resulting in unacceptable reconstructions. Similarly, For BokuHaSitatakaKun from Manga109, compared to the other methods evaluated, none of the models were able to reliably recover a clear and accurate representation of the complex letter M.

### 4.4. Remote Sensing Image Super-Resolution

In comparison to conventional images, remote sensing images exhibit numerous small targets and complex backgrounds, which place higher demands on image super-resolution algorithms. To validate the effectiveness of the proposed SCAN-light, experiments were conducted on publicly available remote sensing datasets, DIOR [71] and DOTA [72].

Two hundred images were randomly selected from the DIOR dataset, and another two hundred images were chosen from the DOTA dataset for conducting the transfer of learning on the proposed model. Sixty images were extracted from the test datasets of both the DIOR and DOTA datasets to evaluate the performance of the proposed SCAN-light.

**Quantitative comparisons.** In order to assess the performance of the proposed SCAN on remote sensing datasets, a comparison is conducted with state-of-the-art lightweight SR methods, including Bicubic, RFDN [24], LAPAR-A [53], IMDN [23], SwinIR-light [45], and MAN-light [42]. Table 3 shows the quantitative comparison on benchmark datasets for upscale factors of ×2, ×3 and ×4. The number of parameters (Params) and floating-point operations (FLOPs) are provided, calculated on the 1280 × 720 output.

Thanks to its efficient architecture, the proposed SCAN model achieves comparable performance while consuming less memory. As shown in Table 3, the proposed approach surpasses all methods in PSNR and SSIM.

Specifically, compared with MAN-light. The proposed SCAN ×4 employs parameters and floating-point operations similar to MAN-light model, achieves an average PSNR improvement of 0.4 dB and an SSIM improvement of 0.0013 on remote sensing datasets.

**Qualitative comparisons.** In addition to the quantitative evaluations, a visual comparison of the proposed SCAN and five state-of-the-art light SR methods are presented, RFDN [24], LAPAR-A [53], IMDN [23], SwinIR-light [45] and MAN-light [42]. Figure 10 presents a visual comparison of state-of-the-art methods on the DIOR and DOTA datasets for up-scale factors of ×4. The images within the red box are cropped and zoomed in on.

For img05916 from DIOR, the proposed SCAN demonstrates the ability to restore clear road surfaces and car contours, a feat that other methods struggle to achieve due to the small size of these regions. Similarly, for imgP0047 from DOTA, the proposed SCAN excels in restoring accurate car contours, while other methods often produce indistinct contours that are difficult to recognize as cars.

### 4.5. Ablation Studies

In this section, we conduct ablation studies on some of the designs involved in our SCAN.

**Study on TSSA.** As previously discussed in Section 3, the FD module and TSGA module are utilized to aggregate triple-scale context information. To substantiate the effectiveness of the FD and TSGA modules, experiments were conducted by selectively removing either of the two from the SCAN-tiny model, and subsequently observing the resultant impact on model performance. As shown in Table 4, it is evident that employing any of them lead to an improvement in performance.

For further demonstration, this is depicted in Figure 11, with the feature maps at different stages within the TSSA module of the model being visualized. Notably, following the collaborative deployment of the TSGA and FD modules, a substantial augmentation in feature richness is observed.

**Study on TSGA**. As previously discussed in Section 3, regarding the extraction of multi-scale information, the input is partitioned into three equal portions, and each of these portions is subsequently fed into three depth-wise-dilated convolutional layers of varying scales. To validate the efficacy of this approach, the performance of the triple-scale approach was compared with single-scale approaches of different sizes. The comparison results are presented in Table 5, revealing that employing a triple-scale approach facilitates the attainment of a balance between performance and computational costs.

**Study on Activation Functions.** As previously discussed in Section 3, a sigmoid-weighted linear unit (SiLU) [60] is utilized in the gating branch. In order to authenticate the effectiveness of SiLUs, experiments were conducted in which the SiLU was replaced with a ReLU [73], PReLU [74], and GELU [59] in the proposed SCAN-tiny model, subsequently comparing their respective performances. The comparison results are shown in Table 6, revealing that leveraging SiLU enables the attainment of superior performance with minimal computational costs.

**The Study on CA.** As previously discussed in Section 3, the introduced CA module consists of a 2-layer channel MLP, a 3 × 3 depth-wise convolutional layer, and a multi-scale feature reallocation module. In Table 7, the results of deploying a 3 × 3 depth-wise convolutional layer and an MSFR module on the proposed tiny SCAN are presented. When comparing the base models without a 3 × 3 depth-wise convolutional layer and multi-scale feature reallocation modules, it is evident that the employment of a 3 × 3 depth-wise convolutional layer can enhance performance. Furthermore, it can also be observed that employing MSFR without the use of large convolutional kernels negatively impacts performance. This is attributed to the pooling operation along the channel dimension, which results in the loss of essential information required for effective processing.

For further clarification, this is demonstrated in Figure 12, where feature maps from various stages within the CA module of the model were visualized, revealing a significant increase in feature richness after the collaborative utilization of the DWConv and MSRA modules.

**Study on tail in spatial and channel aggregation groups (SCAG) and deep feature (DF) extraction module.** As an innovation, a 9 × 9 depth-wise-dilated convolutional layer with a dilation ratio of d = 4 was employed in the tail of the Deep Feature (DF) extraction module and the spatial and channel aggregation groups (SCAG) module for the first time.

In Table 8, a comparison was made between the base models without a tail in the spatial and channel aggregation groups (SCAG) and the Deep Feature (DF) extraction module. It was observed that the inclusion of DWDConv (9 × 9, d = 4) as the tail resulted in a notable improvement in performance.

## 5. Conclusions

In this paper, a Spatial and Channel Aggregation Network (SCAN) for lightweight SISR was proposed. SCAN incorporates the Triple-Scale Spatial Aggregation Attention module (TSSA) to aggregate spatial information at multiple scales. Additionally, the Channel Aggregation (CA) module is used to aggregate channel information. Feature reduction operations are applied in both the TSSA and CA modules to encourage the network to focus on the mid-level information which is challenging for deep neural networks to aggregate. The core concept of SCAN is the utilization of large-kernel convolutions and feature reduction strategies to extract intermediate features that are challenging to capture in both spatial and channel dimensions. Moreover, an innovative approach is introduced by using 9 × 9 large convolutional kernels at the end of the attention modules for the first time, aiming to further enlarge the receptive field. As a result, the proposed SCAN achieves highly efficient SR performance on both public benchmark datasets and remote sensing datasets. Extensive experiments demonstrate that, compared to state-of-the-art lightweight SISR methods with similar parameters and FLOPs, the proposed SCAN provides a significant improvement of 0.13 dB in PSNR metric on benchmark dataset and an even more impressive enhancement of 0.4 dB in PSNR on remote sensing datasets. The proposed SCAN provides a 0.13 dB improvement in peak signal-to-noise ratio (PSNR) and a 0.0013 increase in structural similarity (SSIM). Moreover, on remote sensing datasets, SCAN achieves a 0.4 dB improvement in PSNR and a 0.0033 increase in SSIM.

## Figures and Tables

**Figure 1 sensors-23-08213-f001:**
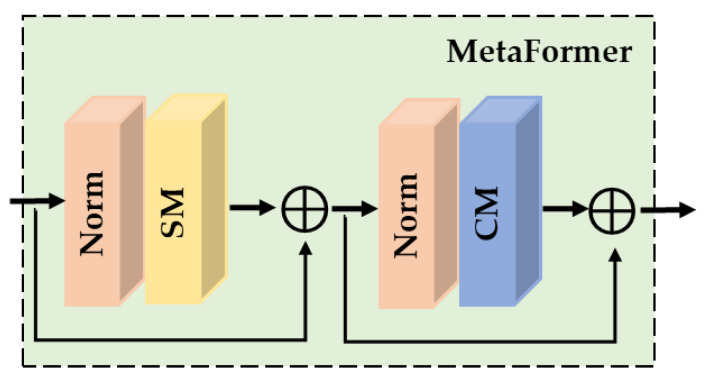
MetaFormer architecture.

**Figure 2 sensors-23-08213-f002:**
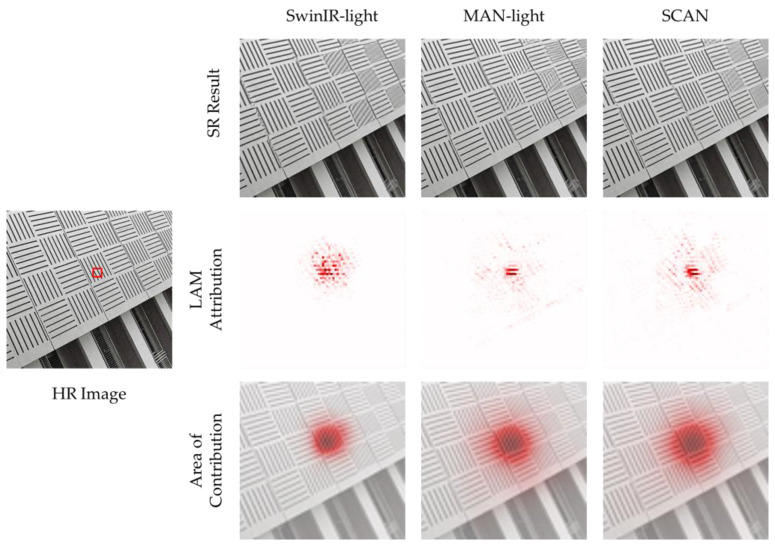
Comparison of local attribution maps (LAMs) [46] between the proposed SCAN and other efficient lightweight SR models. The LAMs demonstrate the significance of every pixel in the input LR image with respect to the patch marked with a red box’s SR. Additionally, the contribution area is shown in the third row. It is evident that the proposed SCAN can aggregate more information.

**Figure 3 sensors-23-08213-f003:**
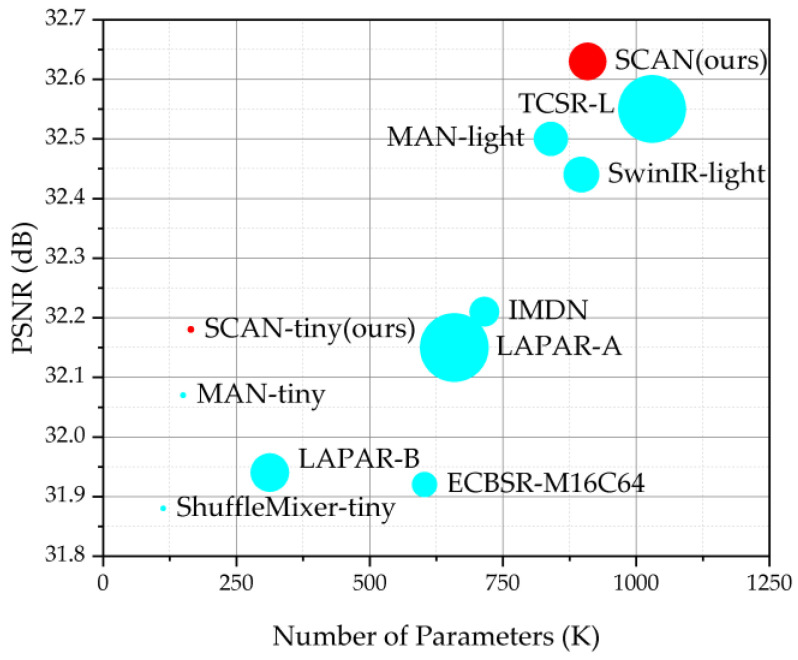
Model performance and complexity comparison between the proposed SCAN model and other lightweight SISR methods on Set5 [47] for ×4 SR. The circle sizes indicate the Floating-Point Operations (FLOPs). The proposed SCAN achieves a good trade-off between model performance and complexity.

**Figure 4 sensors-23-08213-f004:**
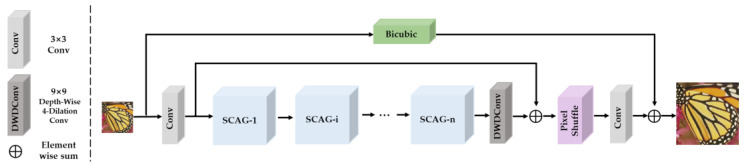
Overview of our spatial and channel aggregation network (SCAN).

**Figure 5 sensors-23-08213-f005:**
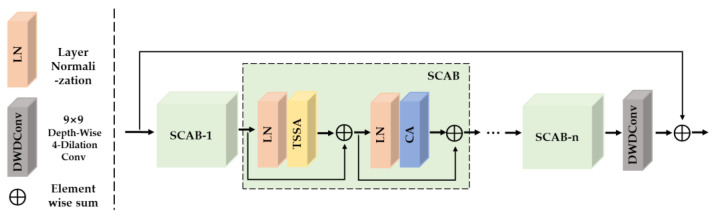
Structure of spatial and channel aggregation groups (SCAG).

**Figure 6 sensors-23-08213-f006:**
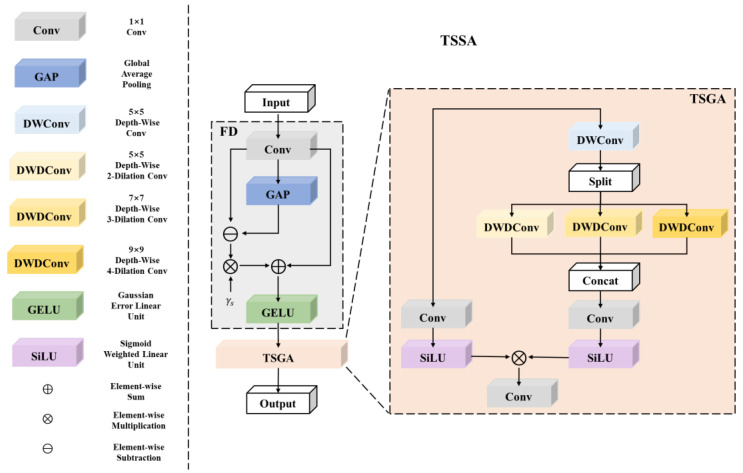
Structure of Triple-Scale Spatial Aggregation (TSSA) module.

**Figure 7 sensors-23-08213-f007:**
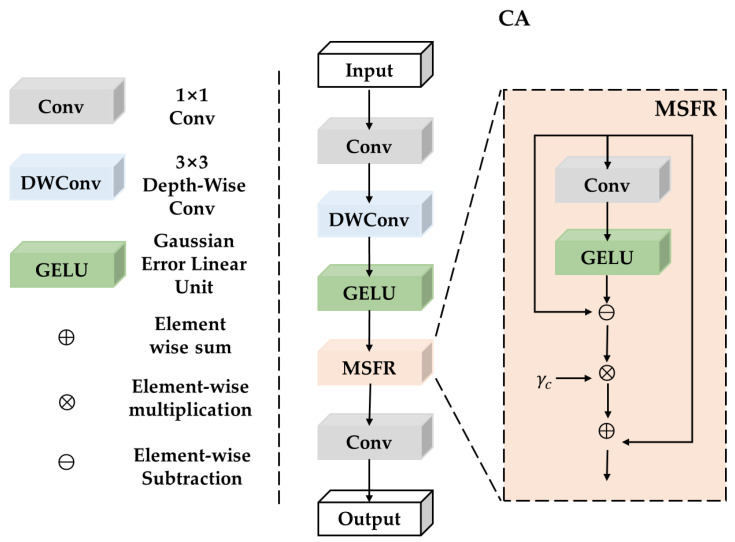
Structure of the Channel Aggregation (CA) module.

**Figure 8 sensors-23-08213-f008:**
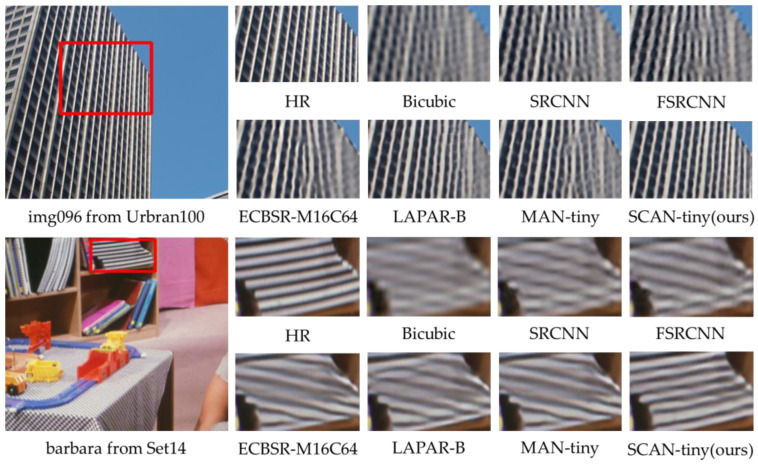
Visual comparison of state-of-the-art methods in some challenging cases (×4SR).

**Figure 9 sensors-23-08213-f009:**
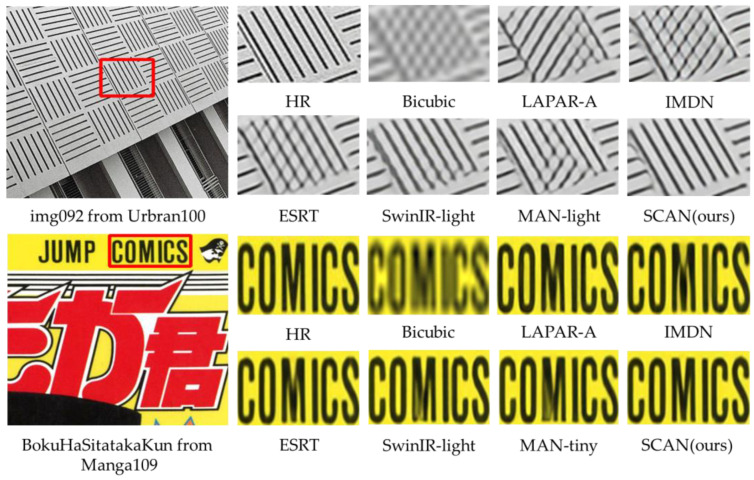
Visual comparison of state-of-the-art methods in some challenging cases(×4SR).

**Figure 10 sensors-23-08213-f010:**
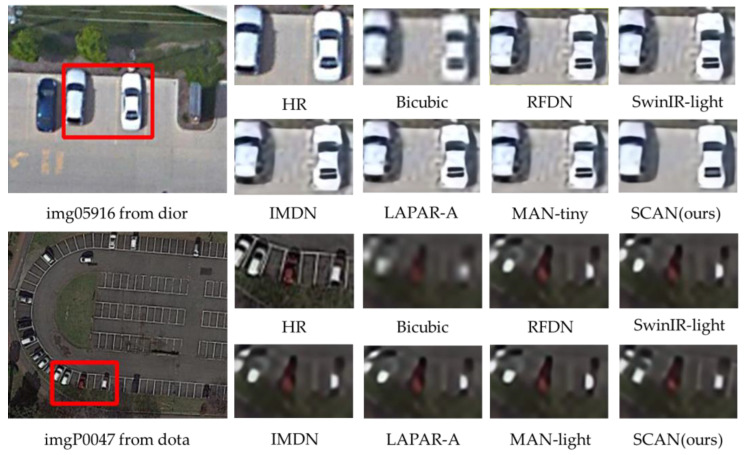
Visual comparison of state-of-the-art methods in some challenging remote sensing cases(×4SR).

**Figure 11 sensors-23-08213-f011:**
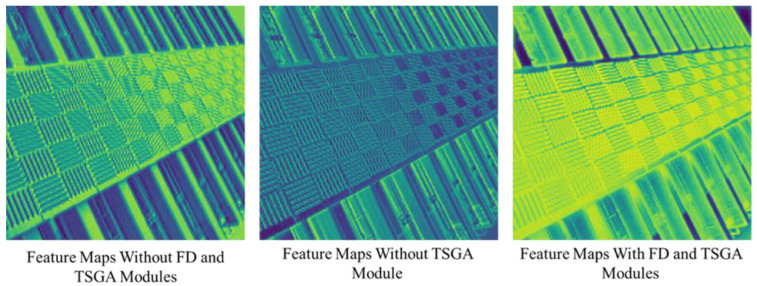
Feature map at the stage of TSSA.

**Figure 12 sensors-23-08213-f012:**
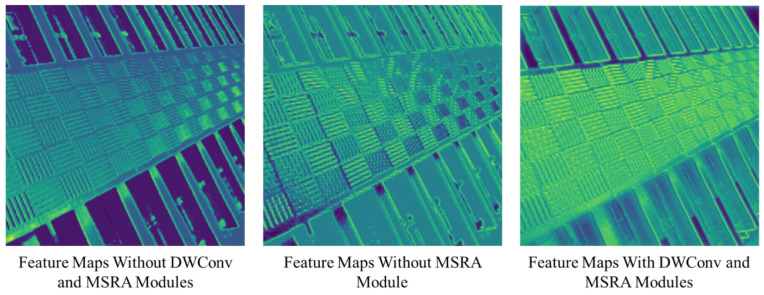
Feature map at the stage of CA.

**Table 1 sensors-23-08213-t001:** Quantitative comparison with state-of-the-art methods for image SR on benchmark datasets. ’Multi-Adds’ is calculated with a 1280 × 720 GT image. The best and second-best performances are in red and blue, respectively.

Method	Scale	Params (K)	Multi-Adds(G)	Set5 PSNR/SSIM	Set14 PSNR/SSIM	BSD100 PSNR/SSIM	Urban100 PSNR/SSIM	Manga109 PSNR/SSIM
Bicubic	×2	-	-	33.66/0.9299	30.24/0.8688	29.56/0.8431	26.88/0.8403	30.80/0.9339
SRCNN [4]		57	53	36.66/0.9542	32.42/0.9063	31.36/0.8879	29.50/0.8946	35.74/0.9661
FSRCNN [20]		13	6.0	37.00/0.9558	32.63/0.9088	31.53/0.8920	29.88/0.9020	36.67/0.9694
ShuffleMixer-tiny [56]		180	25.0	37.85/0.9600	33.33/0.9153	31.99/0.8972	31.22/0.9183	38.25/0.9761
LAPAR-B [53]		250	85.0	37.87/0.9600	33.39/0.9162	32.10/0.8987	31.62/0.9235	38.27/0.9764
MAN-tiny [42]		134	7.7	37.91/0.9603	33.47/0.9170	32.12/0.8991	31.74/0.9247	38.62/0.9770
SAFMN [57]		228	52.0	38.00/0.9605	33.54/0.9177	32.16/0.8995	31.84/0.9256	38.71/0.9771
SCAN-tiny (ours)		150	30.4	37.96/0.9604	33.56/0.9184	32.16/0.8996	31.89/0.9266	38.72/0.9772
Bicubic	×3	-	-	30.39/0.8682	27.55/0.7742	27.21/0.7385	24.46/0.7349	26.95/0.8556
SRCNN [4]		57	53.0	32.75/0.9090	29.28/0.8290	28.41/0.7863	26.24/0.7989	30.59/0.9107
FSRCNN [20]		12	5.0	33.16/0.9140	29.43/0.8242	28.53/0.7910	26.43/0.8080	30.98/0.9212
ShuffleMixer-tiny [56]		114	12.0	34.07/0.9250	30.14/0.8382	28.94/0.8009	27.54/0.8373	33.03/0.9400
LAPAR-B [53]		276	61.0	34.20/0.9256	30.17/0.8387	29.03/0.8032	27.85/0.8459	33.15/0.9417
MAN-tiny [42]		141	8.0	34.23/0.9258	30.25/0.8404	29.03/0.8039	27.85/0.8463	33.32/0.9427
SAFMN [57]		233	23.0	34.34/0.9267	30.33/0.8418	29.08/0.8048	27.95/0.8474	33.52/0.9437
SCAN-tiny (ours)		156	15.8	34.34/0.9265	30.30/0.8414	29.07/0.8043	27.95/0.8483	33.48/0.9436
Bicubic	×4	-	-	28.42/0.8104	26.00/0.7027	25.96/0.6675	23.14/0.6577	24.89/0.7866
SRCNN [4]		57	53.0	30.48/0.8628	27.49/0.7503	26.90/0.7101	24.52/0.7221	27.66/0.8505
FSRCNN [20]		12	4.6	30.71/0.8657	27.59/0.7535	26.98/0.7150	24.62/0.7280	27.90/0.8517
ECBSR-M16C64 [25]		603	34.7	31.87/0.8901	28.39/0.7768	27.44/0.7316	25.63/0.7710	29.80/0.8986
ShuffleMixer-tiny [56]		113	8.0	31.88/0.8912	28.46/0.7779	27.45/0.7313	25.66/0.7690	29.96/0.9006
LAPAR-B [53]		313	53.0	31.94/0.8917	28.46/0.7784	27.52/0.7335	25.84/0.7772	30.03/0.9025
MAN-tiny [42]		150	8.4	32.07/0.8930	28.53/0.7801	27.51/0.7345	25.84/0.7786	30.18/0.9047
SAFMN [57]		240	14.0	32.18/0.8948	28.60/0.7813	27.58/0.7359	25.97/0.7809	30.43/0.9063
SCAN-tiny (ours)		165	9.5	32.18/0.8946	28.59/0.7816	27.55/0.7358	25.97/0.7829	30.34/0.9068

**Table 2 sensors-23-08213-t002:** Quantitative comparison with state-of-the-art methods for image SR on benchmark datasets. ‘FLOPs’ is calculated with a 1280 × 720 GT image. The best and second-best performances are in red and blue, respectively.

Method	Scale	Params (K)	FLOPs (G)	Set5 PSNR/SSIM	Set14 PSNR/SSIM	BSD100 PSNR/SSIM	Urban100 PSNR/SSIM	Manga109 PSNR/SSIM
CARN [21]	×2	1592	222.8	37.76/0.9590	33.52/0.9166	32.09/0.8978	31.92/0.9256	38.36/0.9765
LatticeNet [54]		756	169.5	38.06/0.9607	33.70/0.9187	32.20/0.8999	32.25/0.9288	39.00/0.9774
LAPAR-A [53]		548	171.1	38.01/0.9605	33.62/0.9183	32.19/0.8999	32.10/0.9283	38.67/0.9772
IMDN [23]		694	158.8	38.00/0.9605	33.63/0.9177	32.19/0.8996	32.17/0.9283	38.88/0.9774
ESRT [55]		677	208.4	38.03/0.9600	33.75/0.9184	32.25/0.9001	32.58/0.9318	39.12/0.9774
SwinIR-light [45]		878	195.6	38.14/0.9611	33.86/0.9206	32.31/0.9012	32.76/0.9340	39.12/0.9783
MAN-light [42]		820	180.4	38.18/0.9612	33.93/0.9213	32.36/0.9022	32.92/0.9364	39.44/0.9786
**SCAN (ours)**		889	204.7	38.23/0.9614	34.22/0.9233	32.38/0.9022	33.05/0.9368	39.50/0.9788
CARN [21]	×3	1592	118.8	32.29/0.9255	30.29/0.8407	29.06/0.8034	28.06/0.8493	33.50/0.9440
LatticeNet [54]		765	76.3	34.40/0.9272	30.32/0.8416	29.10/0.8049	28.19/0.8513	33.66/0.9440
LAPAR-A [53]		544	114.1	34.36/0.9267	30.34/0.8421	29.11/0.8054	28.15/0.8523	33.51/0.9441
IMDN [23]		703	71.5	34.36/0.9270	30.32/0.8417	29.09/0.8046	28.17/0.8519	33.61/0.9445
ESRT [55]		770	102.3	34.42/0.9268	30.43/0.8433	29.15/0.8063	28.46/0.8574	33.95/0.9455
SwinIR-light [45]		886	87.2	34.62/0.9289	30.54/0.8463	29.20/0.8082	28.66/0.8624	33.89/0.9464
MAN-light [42]		829	82.5	34.65/0.9292	30.60/0.8476	29.29/0.8101	28.87/0.8671	34.40/0.9434
**SCAN (ours)**		897	91.1	34.81/0.9302	30.65/0.8486	29.31/0.8107	28.80/0.8691	34.62/0.9505
CARN [21]	×4	1592	90.9	32.13/0.8937	28.60/0.7806	27.58/0.7349	26.07/0.7837	30.42/0.9070
LatticeNet [54]		777	43.6	32.18/0.8943	28.61/0.7812	27.57/0.7355	26.14/0.7844	30.46/0.9601
LAPAR-A [53]		659	94.0	32.15/0.8944	28.61/0.7818	27.61/0.7366	26.14/0.7871	30.42/0.9074
IMDN [23]		715	40.9	32.21/0.8948	28.58/0.7811	27.56/0.7353	26.04/0.7838	30.45/0.9075
ESRT [55]		777	67.7	32.19/0.8947	28.69/0.7833	27.69/0.7379	26.39/0.7962	30.75/0.9100
SwinIR-light [45]		897	49.6	32.44/0.8976	28.77/0.7858	27.69/0.7406	26.47/0.7980	30.92/0.9151
TCSR-L [58]		1030	93.0	32.55/0.8992	28.89/0.7886	27.75/0.7423	26.67/0.8039	31.17/0.9107
MAN-light [42]		840	47.1	32.50/0.8988	28.87/0.7885	27.77/0.7429	26.70/0.8052	31.25/0.9170
**SCAN (ours)**		909	51.8	32.63/0.9001	28.91/0.7890	27.80/0.7429	26.79/0.8065	31.38/0.9148

**Table 3 sensors-23-08213-t003:** Quantitative comparison with state-of-the-art methods for light image SR on remote sensing datasets. ‘FLOPs’ is calculated with a 1280 × 720 GT image. The best and second-best performances are in red and blue, respectively.

Method	Scale	Params (K)	FLOPs (G)	Dior PSNR/SSIM	Dota PSNR/SSIM
RFDN [24]	×4	555	23.9	25.95/0.6430	24.79/0.5975
SwinIR-light [45]		840	47.1	27.50/0.6786	26.00/0.7400
IMDN [23]		703	71.5	27.52/0.6778	26.04/0.7381
LAPAR-A [53]		659	94.0	27.55/0.6799	26.01/0.7402
MAN-light [42]		840	47.1	27.59/0.6805	26.19/0.7453
**SCAN (ours)**		909	51.8	28.29/0.7037	26.30/0.7486

**Table 4 sensors-23-08213-t004:** Ablation studies on components of TSSA. The impact of FD and TSGA modules are shown upon SCAN-tiny on the ×4 SR task. ’FLOPs’ is calculated with a 1280 × 720 GT image. The best metrics are highlighted in bold for emphasis.

Module		Params (K)	FLOPs (G)	Set5	Set14	BSD100	Urban100	Manga109
FD	TSGA			PSNR/SSIM	PSNR/SSIM	PSNR/SSIM	PSNR/SSIM	PSNR/SSIM
×	×	99.27	5.63	30.79/0.8654	27.62/0.7584	26.91/0.7178	24.67/0.7359	28.12/0.8610
×	√	153.43	8.75	32.01/0.8922	28.48/0.7786	27.48/0.7332	25.75/0.7755	30.05/0.9032
√	√	165.43	9.45	**32.13/0.8937**	**28.53/0.7800**	**27.52/0.7344**	**25.87/0.7790**	**30.17/0.9046**

**Table 5 sensors-23-08213-t005:** Ablation studies on design of TSGA module. The impact of multi-scale approach, and single-scale approach are shown upon SCAN-tiny on ×4 SR task. ’FLOPs’ is calculated with a 1280 × 720 GT image. The best metrics is highlighted in bold for emphasis.

Convolution Type	Params (K)	FLOPs (G)	Set5 PSNR/SSIM	Set14 PSNR/SSIM	BSD100 PSNR/SSIM	Urban100 PSNR/SSIM	Manga109 PSNR/SSIM
$DWDConv_{5\times 5,d=2}$	159.03	9.08	32.08/0.8931	28.49/0.7792	27.52/0.7339	25.82/0.7774	30.15/0.9043
$DWDConv_{7\times 7,d=3}$	164.79	9.41	32.11/0.8933	28.51/0.7795	27.53/0.7344	25.86/0.7786	30.21/0.9050
$DWDConv_{9\times 9,d=4}$	172.47	9.85	32.10/0.8936	28.52/0.7798	**27.52/0.7346**	**25.88/0.7797**	**30.18/0.9048**
Triple-scale	165.43	9.45	**32.13/0.8937**	**28.53/0.7800**	**27.52**/0.7344	25.87/0.7790	30.17/0.9046

**Table 6 sensors-23-08213-t006:** Ablation studies on different type of activation function. The impact of SiLU, ReLU, PReLU and GELU are shown upon SCAN-tiny on ×4 SR task. ’FLOPs’ is calculated with a 1280 × 720 GT image. The best metrics are highlighted in bold for emphasis.

Activation Function Type	Params (K)	FLOPs (G)	Set5 PSNR/SSIM	Set14 PSNR/SSIM	BSD100 PSNR/SSIM	Urban100 PSNR/SSIM	Manga109 PSNR/SSIM
ReLU	165.43	9.47	32.06/0.7796	28.52/0.7796	27.51/0.7339	25.84/0.7779	30.14/0.9041
PReLU	165.43	9.47	32.09/0.8932	**28.54**/0.7798	27.51/0.7341	25.86/0.7788	30.15/0.9045
GELU	165.43	9.45	32.09/0.8933	28.53/0.7796	27.51/0.7341	**25.87**/0.7786	**30.17/0.9048**
SiLU	165.43	9.45	**32.13/0.8937**	28.53/**0.7800**	**27.52/0.7344**	**25.87/0.7790**	**30.17**/0.9046

**Table 7 sensors-23-08213-t007:** Ablation studies on components of CA. The impact of $DWConv_{3\times 3}$ and MSFR mechanism, are shown upon SCAN-tiny on ×4 SR task. ’FLOPs’ is calculated with a 1280 × 720 GT image. The best metrics are highlighted in bold for emphasis.

Module		Params (K)	FLOPs (G)	Set5	Set14	BSD100	Urban100	Manga109
$DWConv_{3\times 3}$	MSFR			PSNR/SSIM	PSNR/SSIM	PSNR/SSIM	PSNR/SSIM	PSNR/SSIM
×	×	160.15	9.14	32.03/0.8926	28.47/0.7785	27.49/0.7331	25.76/0.7756	30.03/0.9025
×	√	160.63	9.17	31.35/0.8767	27.89/0.7597	27.11/0.7109	24.97/0.7400	28.46/0.8563
√	×	164.46	9.42	32.08/0.8933	28.52/0.7797	27.51/0.7342	25.86/0.7789	30.17/0.9045
√	√	165.43	9.45	**32.13/0.8937**	**28.53/0.7800**	**27.52/0.7344**	**25.87/0.7790**	**30.17/0.9046**

**Table 8 sensors-23-08213-t008:** Ablation studies on different type of tail. The impact of with no tail, $DWDConv_{5\times 5,d=2}$, $DWDConv_{7\times 7,d=3}$ and $DWDConv_{9\times 9,d=4}$ are shown upon SCAN-tiny on ×4 SR task. ’FLOPs’ is calculated with a 1280 × 720 GT image. The best metrics are highlighted in bold for emphasis.

Tail Type	Params (K)	FLOPs (G)	Set5 PSNR/SSIM	Set14 PSNR/SSIM	BSD100 PSNR/SSIM	Urban100 PSNR/SSIM	Manga109 PSNR/SSIM
None	141.81	8.09	31.96/0.8915	28.44/0.7783	27.48/0.7329	25.72/0.7741	29.99/0.9020
$DWDConv_{5\times 5,d=2}$	149.30	8.52	32.09/0.8936	28.52/0.7796	27.52/0.7343	25.88/0.7792	30.20/0.9050
$DWDConv_{7\times 7,d=3}$	156.21	8.91	32.08/0.7796	28.51/0.7796	**27.52/0.7344**	25.85/0.7786	**30.18/0.9048**
$DWDConv_{9\times 9,d=4}$	165.43	9.45	**32.13/0.8937**	**28.53/0.7800**	**27.52/0.7344**	**25.87/0.7790**	30.17/0.9046

## Data Availability

The super-resolution algorithm codes are available online at: https://github.com/linzezuo/SCAN (accessed on 31 Augest 2023).

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
