# Peer review of "Spatial and Channel Aggregation Network for Lightweight Image Super-Resolution"

_sensors, 2023, doi:10.3390/s23198213_

Round 1

Reviewer 1 Report

Comments

In the current manuscript “Spatial and channel aggregation network for lightweight image super-resolution”, the author has proposed a  spatial and channel aggregation network design for image super-resolution, which is composed of cascaded spatial and channel aggregation groups modules. This design may enable the network to focus more on challenging intermediate-level information extraction, leading to improved performance and efficiency of the network. The author also claimed to introduce an innovative 9×9 large kernel convolution to further expand the receptive field.  The research work is interesting and can be used to improve the quality of the images. However, the manuscript is too long and cannot be recommended for acceptance in its current form. The manuscript can be considered for acceptance after it is concise and well-organized.

1.               The abstract is too long. The author should revise and make it more precise using professional and technical terms. The author should provide more details and values of the key achievement parameters for this research work.

2.               English and grammar mistakes should be improved. The complete word for the abbreviations should be mentioned in the first place.

3.               The introduction is extra long with irrelevant information. The author should reduce its content and only add the information necessary to build up the manuscript baseline by adding the attraction, application, progress, problems, and solution provided in this research article.  The last paragraph of the introduction should explain briefly how your work will add innovation. The size of the paragraphs should be almost the same for the whole manuscript. The author should include the latest information about progress in high-resolution imaging and applications such as https://doi.org/10.1002/adom.202300910 to highlight the development in imaging technologies.

4.               Please remove irrelevant references and citations.

5.               Please redraw the graphs using the origin software and make the graph lines bold, clear, and colorful.

6.               The conclusion should be concise with compact findings in this research work.

English and grammar mistakes should be improved.

Author Response

Dear Reviewer,

We extend our heartfelt gratitude for your efforts in reviewing this revised manuscript and providing us with your feedback. Your constructive comments have been instrumental in enhancing the quality of our work.

To facilitate a clear and comprehensive review of the revisions, we have organized this response letter (attached PDF file) in an itemized format. The comments you provided are presented in blue, and our responses are indicated in red. Furthermore, we have highlighted the modifications in the revised manuscript for your convenience. Deletions are denoted by blue centerlined text, while newly added content is highlighted in yellow.

We trust that the modifications made will serve to clarify our ideas and facilitate a better understanding for our readers.

Thank you once again for your valuable comments.

Best Regards,

Feng Huang

Reviewer 2 Report

The paper presents the spatial and channel aggregation network for lightweight Im-age super-resolution. The paper is interesting, however some matters need to be addressed before publication.

1. A better explanation of your contribution needs to be added in the Introduction Section. Please set differences of your work with respect to previous work. Please highligh the advantages, benefits and contributions of the new tecnique and results.

2. Provide more details of the methodology used.

3. It is needed a better description of the results. Please highligh the benefits, the advantages and contributions of your proposal.

Author Response

Dear Reviewer,

We extend our heartfelt gratitude for your efforts in reviewing this revised manuscript and providing us with your valuable feedback. Your constructive comments have been instrumental in enhancing the quality of our work.

To facilitate a clear and comprehensive review of the revisions, we have organized this response letter (attached PDF file) in an itemized format . The comments you provided are presented in blue, and our responses are indicated in red. Furthermore, we have highlighted the modifications in the revised manuscript for your convenience. Deletions are denoted by blue centerlined text, while newly added content is highlighted in yellow.

Once again thank you for your careful reading and constructive comments. Your comments are really helpful to improve the paper. Hope the modifications can clarify our idea and help the reader.

Best Regards,

Feng Huang

Reviewer 3 Report

The authors have devised a low computing cost strategy for the single image super-resolution (SISR) challenge. The SCAN  is a novel experimental approach that aims to retrieve intermediate-level information through the utilization of large kernel convolution. By conducting rigorous trials on widely used photos and satellite data, the suggested methodology demonstrates superior performance compared to state-of-the-art approaches. The ablation study further solidifies the findings.

The research background and literature review exhibit a comprehensive analysis and establish a coherent logical progression in order to effectively convey the significance of this study. The authors have made their source code public, ensuring that the presentation of the suggested method is clear and readily reproducible. Additionally, the results section demonstrates that the suggested method exhibits superior performance compared to SOTAs in a majority of scenarios.

In summary, it is my contention that this manuscript is prepared for dissemination to readers, with the exception of a few errors that require rectification. Please thoroughly proofread this content to eliminate any typographical errors. When encountering the term "triple-scale," it is important to observe that no space should be inserted between the words "triple" and "-scale" (e.g., line 293, 300).

Please thoroughly proofread this content to eliminate any typos.

Author Response

Dear Reviewer,

We extend our heartfelt gratitude for your efforts in reviewing this revised manuscript and providing us with your valuable feedback. Your constructive comments have been instrumental in enhancing the quality of our work.

To facilitate a clear and comprehensive review of the revisions, we have organized this response letter (attached PDF file) in an itemized format. The comments you provided are presented in blue, and our responses are indicated in red. Furthermore, we have highlighted the modifications in the revised manuscript for your convenience. Deletions are denoted by blue centerlined text, while newly added content is highlighted in yellow.

Thank you again for your valuable comments.

Best Regards,

Feng Huang

Reviewer 4 Report

This paper introduces the triple-scale aggregation attention model (TSSA) and proposes spatial and channel aggregation for aggregating both multi-scale and channel information. Moreover, the spatial and channel aggregation network (SCAN), a lightweight and efficient pure CNN-based SISR network model that combines the advantages of both CNN and transformer, was proposed, and quantitative and qualitative evaluations were conducted on benchmark datasets and remote sensing datasets to investigate the proposed SCAN. The contribution of this paper is prominent. However, there are several flaws in this paper, as follows.

 1.     The abstract should summarize the research goal, the principal results, significant conclusions, and primary research outcomes. An abstract is typically presented independently from the article. Therefore, it must be stand-alone. Please modify the abstract considering all those mentioned above.

2.     Some references need to give specific information, such as 19, 22,23, 35, 36,38,45,47, 49, 50, 52, 54, 58-62, 69, 72, 82.

Moderate editing of English language required.

Author Response

(The authors gave the same response as above.)

Round 2

Reviewer 1 Report

Manuscript is well revised and can be published.

Please recheck all English and grammar mistakes during proof read process.